# A Timeline of Biosynthetic Gene Cluster Discovery in *Aspergillus fumigatus*: From Characterization to Future Perspectives

**DOI:** 10.3390/jof10040266

**Published:** 2024-04-02

**Authors:** Hye-Won Seo, Natalia S. Wassano, Mira Syahfriena Amir Rawa, Grant R. Nickles, André Damasio, Nancy P. Keller

**Affiliations:** 1Department of Medical Microbiology and Immunology, University of Wisconsin, Madison, WI 53706, USA; hseo45@wisc.edu (H.-W.S.); wassano@wisc.edu (N.S.W.); bintiamirraw@wisc.edu (M.S.A.R.); gnickles@wisc.edu (G.R.N.); 2Department of Biochemistry and Tissue Biology, Institute of Biology, University of Campinas (UNICAMP), São Paulo 13083-970, Brazil; adamasio@unicamp.br; 3Department of Plant Pathology, University of Wisconsin, Madison, WI 53706, USA

**Keywords:** *Aspergillus fumigatus*, secondary metabolite, natural product, biosynthetic gene cluster

## Abstract

In 1999, the first biosynthetic gene cluster (BGC), synthesizing the virulence factor DHN melanin, was characterized in *Aspergillus fumigatus*. Since then, 19 additional BGCs have been linked to specific secondary metabolites (SMs) in this species. Here, we provide a comprehensive timeline of *A. fumigatus* BGC discovery and find that initial advances centered around the commonly expressed SMs where chemical structure informed rationale identification of the producing BGC (e.g., gliotoxin, fumigaclavine, fumitremorgin, pseurotin A, helvolic acid, fumiquinazoline). Further advances followed the transcriptional profiling of a Δl*aeA* mutant, which aided in the identification of endocrocin, fumagillin, hexadehydroastechrome, trypacidin, and fumisoquin BGCs. These SMs and their precursors are the commonly produced metabolites in most *A. fumigatus* studies. Characterization of other BGC/SM pairs required additional efforts, such as induction treatments, including co-culture with bacteria (fumicycline/neosartoricin, fumigermin) or growth under copper starvation (fumivaline, fumicicolin). Finally, four BGC/SM pairs were discovered via overexpression technologies, including the use of heterologous hosts (fumicycline/neosartoricin, fumihopaside, sphingofungin, and sartorypyrone). Initial analysis of the two most studied *A. fumigatus* isolates, Af293 and A1160, suggested that both harbored ca. 34–36 BGCs. An examination of 264 available genomes of *A. fumigatus* located only four additional new BGCs, suggesting the secondary metabolome across *A. fumigatus* isolates is remarkably conserved. Based on our analysis, around 20 of the genetically characterized BGCs within the *A. fumigatus* species complex still lack a known chemical product. Such BGCs remain the final hurdle in fully understanding the secondary metabolism in this important species.

## 1. Introduction

Filamentous fungi are renowned for the synthesis of small bioactive compounds commonly called secondary metabolites (SMs) or natural products. Although the biological role of SMs in producing fungus was originally discarded or, at best, unknown, studies in the last two decades have shown that SMs were critical fitness factors for fungal success in varied environments. For example, in considering the opportunistic pathogen *A. fumigatus* alone, its SMs play various roles [1] in defending against or killing host cells [2], acquiring essential micronutrients [3], mediating interacts with other microorganisms [4,5], and protection from environmental extremes such as UV radiation [6].

Each fungal SM is typically synthesized by a specific biosynthetic gene cluster (BGC), where genes required for the SM are co-regulated and arranged contiguously in a locus [7]. The current understanding of BGCs was only realized with genome sequencing of filamentous fungi, first observed in *Aspergillus* spp. [8]. Depending on genus and species, the number of BGCs in any fungus may range from 0 to 80, with filamentous Ascomycetes containing, on average, the highest numbers [9]. In most fungi, the products of their BGCs are unknown but are of interest due to potential useful bioactivities. Several *Aspergillus* spp. stand out as having the most defined BGCs, with *A. fumigatus* and *A. nidulans* having the highest percentage of chemically defined BGC products [10,11].

Efforts to identify products of *A. fumigatus* BGCs first centered on those metabolites commonly expressed in growth media and deemed to be important in invasive aspergillosis, such as DHN-melanin, gliotoxin, and fumitremorgin [12]. However, many *A. fumigatus* BGCs and their products were found to be silent in standard laboratory conditions, and later studies employed either endogenous overexpression strategies, the use of heterologous hosts, or more creative growth conditions such as co-culture with bacteria to activate these BGCs. With many efforts from multiple laboratories, 20 *A. fumigatus* BGCs have now been defined.

Here, we present a historical journey of when each of these 20 BGCs was linked to their products. We posit that the current commonly used *A. fumigatus* isolates, Af293 and A1163, are unlikely to yield new compounds without considerable genetic manipulation and demonstrate that common parameters known to impact SM production, such as an epigenetic mutation or changes in temperature and media composition, primarily alter the titers of known SMs. However, an analysis of BGC composition across 264 sequenced isolates of *A. fumigatus* suggests that new BGC products may be obtained from diverse strains of this fungus.

## 2. Materials and Methods

### 2.1. Strains and Culture Conditions

The *A. fumigatus* strains used in this study type (A1160 and Δ*sirE* [13]) were maintained in glycerol stocks and activated on glucose minimal medium GMM [14] with added 0.12% (*w*/*v*) uracil/uridine. Czapek yeast extract agar (CYA) (Thermo Fisher Scientific, Waltham, MA, USA), regular GMM, and GMM replacing the nitrogen source NaNO_3_ to NH_4_Cl were used as the 3 media for secondary metabolite extraction.

### 2.2. Secondary Metabolite Extraction and LC-MS/MS Analysis

For metabolomic analysis, 1 × 10^5^ spores were point-inoculated on GMM, CYA, and NH_4_Cl agar in triplicates and cultivated at 37 °C for 7 days and at 25 °C for 14 days. After culturing, the total contents of each Petri dish were freeze-dried, lyophilized, and extracted with 30 mL methanol (Sigma Aldrich, St. Louis, MO, USA). The supernatant was then vacuum-filtered, and the solvent was removed under reduced pressure. The same procedure was performed for the control culture media. The final extracts were stored at −20 °C. For sample preparation, HPLC-grade methanol was added to reach 5 mg/mL for each extract, and the samples were sonicated until complete dissolution. LC−MS was performed on a Thermo Fisher Scientific- Vanquish UHPLC system coupled with a Thermo Q-Exactive HF hybrid quadrupole-orbitrap high-resolution mass spectrometer equipped with a HESI ion source (Thermo Fisher Scientific, Waltham, MA, USA). Each sample was analyzed in negative and positive ionization modes using an *m*/*z* range of 100 to 1500. A Waters XBridge BEH-C18 column (2.1 × 100 mm, 1.7 μm) (Waters, Milford, MA, USA) was used with 0.1% formic acid in acetonitrile (organic phase) and 0.1% formic acid in water (aqueous phase) as solvents at a flow rate of 0.2 mL/min. An amount of 5 μL of samples was injected. A 20-minute solvent gradient scheme was used, starting at 5% organic with a linear increase to 98% for 10 min, holding at 98% organic for 5 min, decreasing back, and holding at 5% organic for 5 min for a total of 20 min.

### 2.3. Feature Detection and Characterization

LC−MS RAW files were converted to mzXML format (centroid mode) using RawConverter (v.1.2.0.1) (The Scripps Research Institute, San Diego, CA, USA) followed by analysis using the XCMS analysis. Positive and negative ionization mode data were processed separately. XCMS features with *p*-values > 0.05 were filtered out, and the volcano plots were created using Excel (2024). The intensities of *m*/*z* values of known SMs were extracted out of mass spectra, and graphs were plotted using GraphPad Prism version 10.2.0.

### 2.4. Dataset Generation and Secondary Metabolite Annotation

All annotated and publicly available *A. fumigatus* genomes (264 genomes) were downloaded from the NCBI database on 1 December 2022 using the NCBI’s Dataset tool (v14.15.0). To generate the BGC predictions, we ran fungal antiSMASH (v6; default settings) [15] on every *A. fumigatus* genome. For all 19 known clusters, cBlaster (v1.3.18) [16] was utilized to check for homologous loci across the other *A. fumigatus* genomes. All antiSMASH predictions can be found in this paper’s to Appendix A.

### 2.5. Grouping the A. fumigatus BGC Predictions into Gene Cluster Families

Homologous BGCs are thought to produce identical or closely related SMs and are referred to as gene cluster families (GCFs) [17]. To determine which of our detected BGCs were members of shared GCFs, all antiSMASH predictions were run through BiG-SCAPE (e.g., Biosynthetic Gene Similarity Clustering and Prospecting Engine; v1.1.5) [18]. A total of seven BiG-SCAPE cutoff values between 0.1 and 0.7 by increments of 0.1 were tested. Values greater than 0.5 were found to be too relaxed, leading to the major merging of large GCFs, which were separated at lower cutoffs. In the end, an optimal cutoff value of 0.3 (which is also the default value) was chosen for generating initial GCF classifications. A network visualization was created with Cytoscape (v3.9.1) [19] for each natural product class and can be seen in Appendix A. Each GCF generated by BiG-SCAPE was manually inspected to confirm that its genetic components did not resemble any known BGCs. This re-evaluation uncovered that some originally “rare” GCFs were products of isolate-to-isolate discrepancies in orthologous BGC boundaries identified by antiSMASH. Furthermore, it was observed that BGCs exhibiting features of various biosynthetic types often corresponded to GCFs spanning multiple biosynthetic classes. A plot visualizing the four GCFs that we believe are novel to this study can be found in Figure 6.

### 2.6. Creating the Estimated Aspergillus fumigatus Species Complex Phylogeny

We employed a coalescent model-based approach to construct the phylogeny from 264 *A. fumigatus* genomes due to this method’s scalability and ability to handle missing data [20]. Two outgroup species, *Aspergillus lentulus* and *Aspergillus fischeri*, were added to the dataset based on their known close relation to *Aspergillus fumigatus* [21]. All gene trees were made using the following pipeline. BUSCOs (e.g., ‘Benchmarking Single Copy Orthologs’; 4194 genes in total) were identified using Orthofinder v2.5.2 (default settings) [22,23]. A smaller set of 700 highly conserved BUSCOs was filtered from the larger database for computational scalability. These 700 shared BUSCOs were aligned using MAFFT (v7.520) with the ‘-auto’ parameter [24]. All alignments were trimmed using trimAl (v1.2) and the ‘-gappyout’ parameter [25]. The optimal model of sequence evolution was chosen using the built-in version of ModelFinder found in IQ-Tree (v2.2.0) [26]. Lastly, the phylogenic trees were constructed using IQ-Tree [27] and run with 1000 ultrafast bootstrap replicates. The resulting 700 gene trees were used as the input for ASTRAL (e.g., Accurate Species TRee Algorithm; v5.7.8) [28], which was run with default settings. All trees created for this publication can be found in Appendix A.

## 3. Results

### 3.1. Timeline of Linkage of Natural Products to Specific BGCs

Dihydroxynapthalene (DHN) melanin was the first metabolite to be linked to a specific BGC (Figure 1 and Figure 2, Appendix A). DHN-melanin is a negatively charged, hydrophobic pigment that coats the asexual spore cell wall, and similar pigments are found in plant pathogenic fungi [29]. A requirement for this pigment for full virulence in *A. fumigatus* was first reported in 1997 [30]. The relationship of melanin biosynthesis across fungi was noted when it was observed that the agricultural fungicide tricyclazole inhibited the conidial pigmentation of *A. fumigatus* [31]. The first gene encoding a polyketide synthase, *pksP/alb1*, was first identified in 1998 [32], and the whole BGC consisting of six genes was discovered in 1999 [33]. Since then, scores of papers have been published on the role of DHN-melanin in virulence [34], as well as a UV protective agent [6].

Three BGCs were identified in 2005, one being the gliotoxin (GT) BGC (*gli* BGC) (Figure 1 and Figure 2, Appendix A). GT was first detected from *A. fumigatus* in 1944 [35]. GT is a toxic epidithiodioxopiperazine due to a disulfide bridge in its structure that is responsible for generating reactive oxygen species (ROS) [36,37]. The *gli* BGC was identified via a homology search using the sirodesmin BGC, sirodesmin being a similarly structured epidithiodioxopiperazine produced by the plant pathogenic fungus *Leptosphaeria maculans* [38,39]. The *gli* BGC consists of twelve genes, including the positive-acting transcription factor, GliZ [40]. The toxic nature of GT provides a protective shield for *A. fumigatus* not only against immune cells during pathogenesis [41] but also against amoebae and bacteria in the environment [42,43].

The fumigaclavine BGC (*fga/eas* BGC) (Figure 1 and Figure 2, Appendix A) shared a similar discovery history as gliotoxin in the sense that the BGC was found via similarities to a known ergot alkaloid BGC in the plant pathogen *Claviceps purpurea* in 2005 [44,45]. Biosynthetic studies took a bit longer to complete where at least two genes, *pesL* and *pes1*, were suggested to be necessary for fumigaclavine C synthesis in 2012 [46], but a fuller characterization of the BGC was not completed until 2020 [47]. Fumigaclavine C synthesis is required for full virulence in an insect model of invasive aspergillosis [48].

The fumitremorgin BGC (*ftm* BGC) (Figure 1 and Figure 2, Appendix A) was also identified in 2005 via similarity to a known BGC from *C. purpurea* [49]. Fumitremorgin is a prenylated indole alkaloid mycotoxin with a diketopiperazine core structure. At first, it was reported that *ftm* BGC had ten genes [50]. But, four years later, in 2009, based on the data provided by genome-mining [51], it was corrected to the fact that *ftm* BGC contained nine *ftm* genes [52]. These types of molecules can inhibit cell cycle progression and demonstrate neurotoxicity [53,54], although no direct study of this metabolite in invasive aspergillosis has been conducted.

The first genome sequence of *A. fumigatus* in 2005 [51] noted an NRPS/PKS hybrid, a type of enzyme found in bacteria [55] but had not been widely reported in fungi at the time [56,57]. In 2007, deletion and overexpression of the encoding gene (later called *psoA/nrps14*) showed it to be involved in the production of pseurotin A [58]. However, it was not until 2013 that the full BGC (*pso* BGC) was characterized (Figure 1 and Figure 2, Appendix A) [59]. In this study, it was found that genes for fumagillin and pseurotin were physically intertwined in a single supercluster. Pseurotin is not described as having a role in aspergillosis, but interestingly, this metabolite has been recently found to exhibit antibacterial properties and mediate the bacterial composition of the cheese rind microbiome [60].

The helvolic acid BGC (*hel* BGC) was characterized in 2009 (Figure 1 and Figure 2, Appendix A) [61]. Helvolic acid, first identified from the *A. fumigatus* in 1943 [62], is a terpene derived mycotoxin [63]. This metabolite is an effective antibacterial agent against Gram-positive bacteria [64,65] that, similarly to pseurotin, appears to mediate fungal domination over bacteria in specific microbiomes [66]. There are no reports of helvolic acid affecting the virulence of *A. fumigatus*, although the molecule has been found to slow ciliary beat frequency in mammalian cell lines, a result speculated to potentially influence colonization of the airways [67].

Fumiquinazoline was first discovered in 1995 (Figure 2) [68], and its BGC (*fmq* BGC) was identified in 2010 (Figure 1, Appendix A) by searching for an anthranilate-activating NRPS in the *A. fumigatus* genome [69]. The *fmq* BGC was expanded to five genes in 2014 [70]. This metabolite selectively accumulates in *A. fumigatus* conidia [70,71], where it provides some protection against UV radiation [6]. Although there are no studies on the potential impact of this metabolite on virulence, one study reported that fumiquinazoline could inhibit phagocytosis by both the soil amoebae *Dictyostelium discoideum* and murine macrophage [72].

In 2010, the BGC responsible for synthesizing pyripyropene A (*pyr* BGC) was identified (Figure 1 and Figure 2, Appendix A) [73]. Pyripyropene A was initially identified as a potent inhibitor of acyl-CoA cholesterol acyltransferase, which is a mammalian intracellular enzyme in the endoplasmic reticulum [74,75]. This metabolite also shows promise as an insecticide, although the mode of action is not fully realized [76]. A recent study shows that insects may evolve resistance to pyripyropene-like insecticides [77].

Another BGC encoding a conidial SM, endocrocin, was identified in 2012 (Figure 1 and Figure 2, Appendix A) [78]. The BGC (*enc* BGC) was identified by searching for products encoded by non-reducing PKS that lack a thioesterase/Claisen cyclase domain. Endocrocin belongs to a common chemical class called anthraquinones, which is noted for its industrial uses [79]. Endocrocin was shown to inhibit neutrophil migration both in vitro, using human neutrophils, and in vivo, using the zebrafish model [80].

Three BGCs were identified and linked to metabolites in 2013 (Figure 1, Appendix A). Fumagillin was identified in 1951 (Figure 2) [81] when it was found to possess amoebicidal activity. This meroterpenoid is a known inhibitor of methionine aminopeptidase 2 [82] and is used as a protectant against Nosema disease of honeybees [83]. As mentioned earlier, the biosynthetic genes encoding for fumagillin synthesis are intertwined with those encoding pseurotin production [59]. The resulting ‘supercluster’ is regulated by the BGC-specific factor, FumR/FapR [59,84]. This supercluster is conserved in the distantly related insect pathogen *Metarhizium*, where, together, the metabolites demonstrate antibacterial activity [85]. Fumagillin is involved in host cellular damage and appears to protect the fungus from phagocytosis [86].

The second BGC, characterized in 2013, encoded the iron-chelating non-ribosomal peptide hexadehydroastechrome (Figure 1 and Figure 2, Appendix A) [87]. Hexadehydroastechrome is a key player in iron homeostasis in *A. fumigatus* [88] and also impacts the expression of several other SMs, such as gliotoxin. Overexpression of hexadehydroastechrome increased the virulence of *A. fumigatus* in a murine model of aspergillosis [87].

The third BGC, characterized in 2013, a silent BGC, synthesized neosartoricin, a prenylated polyphenol (Figure 1 and Figure 2, Appendix A) [89,90]. This metabolite was found the same year by another research group, which called the compound fumicycline [91]. The neosartoricin (or fumicycline) BGC (*nsc/fcc* BGC) is conserved in dermatophytic fungi. In one study, the dermatophyte BGC was heterologously expressed in the model fungus *A. nidulans* [90], and in another study, the pathway-specific TF, NscR/FccR was activated in *A. fumigatus* using the constitutive *gpdA* promoter, which induced expression of the other BGC genes [89]. In the third study, all genes in the *nsc/fcc* BGC were activated, and neosartoricin/fumicycline was produced by co-culturing with *Streptomyces rapamycinicus* [91]. Bioactivity assays showed that the purified metabolite exhibited moderate activity against *S. rapamycinicus*, leading the authors to speculate that neosartoricin/fumicycline may contribute to fungal defense.

A fourth conidial metabolite, trypacidin, was linked to its BGC (*tpc/tyn* BGC) in 2015 by two groups (Figure 1 and Figure 2, Appendix A) [92,93]. Trypacidin was first discovered in 1963, and its antibiotic properties were reported at that time [94]. In addition to its antibiotic and antiprotozoal properties, trypacidin possesses anti-phagocytosis characteristics against macrophages and amoebae [92]. Trypacidin is a spirocoumaranone [95] and actually synthesizes endocrocin as an early precursor in the long trypacidin pathway [93]. Trypacidin is an example of a temperature-regulated natural product [96].

In 2016, a BGC encoding, an unusual core synthase, was characterized [97,98]. The fumisoquin BGC contains an NRPS-like enzyme that lacks a condensation domain found in canonical NRPS (Figure 1 and Figure 2, Appendix A). This BGC was found by searching for orphan BGCs that are under the control of LaeA [99]. Fumisoquin biosynthesis is notable for its carbon-carbon bond formation between two amino acid-derived moieties that are directly analogous to isoquinoline alkaloid biosynthesis in plants, supporting a view of divergent evolution in fungi and plants to synthesize a similar end chemistry [97].

The year 2018 marked the addition of a relatively obscure chemical class to the compendium of characterized *A. fumigatus* natural products: the isocyanides, also known as isonitriles, which are characterized by the bioactive functional group R-N^+^≡C^−^ (Figure 1 and Figure 2, Appendix A) [100]. The first characterized isocyanide compound, xanthocillin, was discovered in 1948 from a culture of *Penicillium notatum* [101]. In 2011, a co-culture between *A. fumigatus* and *Streptomyces peucetius* led to the production of the isocyanide xanthocillin analog BU-4704 [102], and in 2018, a BLAST search targeting bacterial isocyanide synthases (ICS) unearthed four prospective ICS homologs distributed across three BGCs in *A. fumigatus* [100]. Subsequently, it was revealed that one of these ICS BGCs encoded the tyrosine-derived xanthocillin. Overexpression of the BGC TF, *xanC*, greatly increased product synthesis, allowing the discovery of the copper chelating properties of this compound that was related to antibacterial activity [103].

In 2019, fumihopaside A was characterized as a new hopane-type glucoside (Figure 2). Fumihopaside A is synthesized by the four-gene *afum* BGC (Figure 1, Appendix A). This cluster was identified via a search to find natural products with glycosyl modifications [104]. One of the four genes, *afumC*, encodes glycosyltransferase, which was the key to finding this glucoside. The products of the *afum* BGC, fumihopaside A and B, were confirmed both by deleting members of the cluster in *A. fumigatus* as well as expressing the cluster in *A. nidulans*. This study demonstrated that fumihopaside A enhances the thermotolerance and UW resistance of *A. fumigatus*.

Similar to the finding that *S. peucetius* induced BU-4704 synthesis [102] and that *S. rapamycinicus* induced fumicycline A production [91], the α-pyrone polyketide fumigermin, was identified in 2020 (Figure 1 and Figure 2, Appendix A) during *A. fumigatus/S. rapamycinicus* co-culture [105]. Fumigermin was shown to inhibit the germination of *S. rapamycinicus* spores. Co-culturing with other *Streptomyces* spp. such as *S. iranensis*, *S. coelicolor* or *S. lividans* also induced fumigermin production. This finding added to the accruing data showing that some SMs and their BGCs are activated as defense molecules in microbiome settings.

The year 2022 brought the characterization of two more isocyanide products, fumivaline and fumicicolin, both generated from the copper-responsive metabolite (*crm*) BGC consisting of four genes (Figure 1 and Figure 2, Appendix A). The *crm* cluster is activated by low copper concentrations. Within the *crm* BGC, there is a core gene encoding a multi-domain ICS-NRPS-like enzyme called CrmA. This enzyme modifies L-valine into (S)-2-isocyanoisovaleric acid, commonly referred to as valine isocyanide, which serves as a crucial intermediate for the synthesis pathways of both fumivaline and fumicicolin [106]. While fumivaline A and fumicicolin A lack individual antibacterial/antifungal properties, their combined treatment synergistically inhibits the growth of some bacteria and fungi.

In addition to isocyanide products, the sphingofungin BGC was also identified in 2022 (Figure 1 and Figure 2, Appendix A) [107]. Sphingofungin is a polyketide-derived compound, which was first isolated from *A. fumigatus* in 1992 [108]. Sphingofungin is known to inhibit serine palmitoyl transferase (SPT) [109], which plays a crucial role in the biosynthesis of sphingolipids (SLs) [110]. Recently, using confocal microscopy, it was shown that the synthesis of sphingofungins and SLs was partially co-compartmentalized in the ER and ER-derived vesicles [111].

The most recent *A. fumigatus* SM to be linked to a BGC is sartorypyrone (Figure 1 and Figure 2, Appendix A) [112]. Sartorypyrone was produced by heterologously expressing its six gene BGC in a production strain of *A. nidulans*. Like fumicycline A, satorypyrone is a meroterpenoid, a chemical class that is derived from hybrid polyketide or non-polyketide and terpenoid biosynthesis. Sartorypyrones are known metabolites produced by other fungi and have been reported to exhibit antibacterial activity against Gram-positive bacteria, including *S. aureus*, *B. subtilis*, *E. coli*, and *P. aeruginosa* [113,114].

### 3.2. Shifts in Secondary Metabolite Profiles Mediated by Epigenetics, Temperature, and Media Alter Titers of Commonly Expressed Compounds

Many BGC/SM linkage discoveries were in one of two commonly assessed *A. fumigatus* isolates, Af293 or A1163, both of which have fully annotated genomes [115]. As seen by the timeline, the first BGCs to be fully characterized were those associated with the metabolites (e.g., melanin, gliotoxin, fumigaclavine, fumitremorgin, fumagillin, helvolic acid, etc.) produced in standard laboratory conditions. Greater efforts—including co-culture, unusual growth conditions (metal extremes), and molecular manipulations in heterologous hosts—were needed to characterize additional BGCs and their metabolites (e.g., fumihopaside, fumivaline, fumicicolin, fumigermin, sphingofungin, sartorypyrone, etc). Thus, it appears that a specified reduced set of SMs are preferentially synthesized by *A. fumigatus*, at least in isolates Af293 and A1160, unless challenged by more extreme abiotic or biotic factors.

To address this observation, we grew WT and a sirtuin E deletion strain (Δ*sirE*) in different media and temperatures to analyze any shifts in SM production. The Δ*sirE* was chosen as the loss of SirE, a lysine deacetylase, that had previously been reported to impact secondary metabolite profiles [13]. Volcano plots of all conditions demonstrate that each variable—strain, media, and temperature—altered the SM profile (Figure 3). The significant *p*-values were set at <0.05, and log2 fold change was set at 1 and −1 (2-fold higher and lower). Known and putative SMs produced by *A. fumigatus* in different strains were identified based on the analyses via MAVEN, SIRIUS, and XCMS platforms (Figure 3 and Figure 4). About 27 known metabolites were identified and listed in Appendix A, in which their *m*/*z* values were confirmed either by comparisons with standards or via searches on the Pubmed database and previous papers. Their abundance was determined based on the height intensity of *m*/*z* values (precursor ions) displayed on ion chromatograms and mass spectra.

Of the known SMs, only nine showed significant differences across the six treatments: fumigaclavine A; fumigaclavine C; fumitremorgin C; fumagillin; fumiquinazoline A; pyripyropene A; pseurotin A; gliotoxin; and terezine D (the stable precursor to hexadehydroastechrome) (Figure 3 and Figure 4). The overall observation was that the pattern of regulation changed for each condition; e.g., different metabolites were up- or downregulated in the Δ*sirE* strain compared to WT, and this was also dependent on media and temperature. For instance, fumigaclavine A was upregulated in Δ*sirE* in GMM at 37 °C but downregulated at 25 °C. The most noteworthy observation from the entire study was that only those SMs commonly produced in standard laboratory conditions were reliably measurable regardless of treatment.

On the other hand, when analyzing the global features produced in these treatments, there were large differences dependent on strain, temperature, and media. Features (unknown metabolites) were characterized by analytes with specific mass-to-charge (*m*/*z*) values and retention times observed when the samples were subject to LC-MS. Whether or not these features could represent new SMs is unknown. Additionally, principal component analysis (PCA) was performed to visualize the relationships between different strains, media, and temperature conditions. The scores plot (Appendix A) illustrated that *A. fumigatus* WT and Δ*sirE* strains were grouped into five main clusters. Three main clusters were apparently grouped based on media compositions. All strains grown with NH_4_Cl as a nitrogen source were clustered together. The scores plot also suggested that the WT strain was cultivated at different temperatures, but in the same media, it shared the same feature characteristics. The most striking observation was that the Δ*sirE* strain grown at 25 °C in both CYA and GMM was clustered individually and separately from the rest of the growth conditions, suggestive of a different suite of metabolites unique to Δ*sirE* at these temperatures.

### 3.3. Variation in BGCs in A. fumigatus Strains

Could the SM output of *A. fumigatus* vary across different isolates? As mentioned above, almost all BGC characterization has been performed using isolates Af293 or A1160, which have typically been reported to harbor ca. 34–36 BGCs [116,117]. However, recent studies in other fungi have shown a significant variation in BGC composition in different isolates. For example, a recent analysis of 94 isolates of *A. flavus* showed BGCs could be grouped in ‘core’ (all or most strains contained the BGC) and ‘accessory’ BGCs that grouped with phylogenetic populations [118]. These *A. flavus* genotype differences also correlated with chemotypic differences in SM production. Further, previous work revealed as many as five unique variations in BGCs across 66 strains of *A. fumigatus* [117]. These studies indicated that more chemical diversity might exist within the *A. fumigatus* species complex than can be seen by studying Af293 or A1160.

We aimed to reproduce this analysis of the 264 publicly available *A. fumigatus* annotated genomes. The popular genome mining program, antiSMASH [15], was run on every genome to generate initial BGC predictions. For all 22 characterized BGCs, the presence or absence of these clusters in the genomes was manually double-checked using cBlaster [16], a tool that identifies homologous, co-localizing sets of genes. In such cases, we looked for the presence of the entire reference cluster, not only the key backbone synthase/synthetases. Lastly, we ran BiG-SCAPE on all the antiSMASH-generated predictions to organize the BGCs into families of evolutionarily related BGCs. Such families are often referred to as gene cluster families (GCFs) and are thought to produce the same or highly similar classes of SMs. Our generated GCFs were compared to those detected by our group in 2017 [117] to determine what new BGCs could be detected with more modern genome mining algorithms and a larger dataset of *A. fumigatus* genomes.

Figure 5 illustrates the presence and absence of the 22 defined BGCs described above, as well as any uncharacterized GCFs from our analysis that matched those previously identified by Lind et al. (2017) [117]. Overall, we found that SM-producing BGCs within the *A. fumigatus* species complex were remarkably conserved across different populations (Figure 5). The two exceptions to this appeared to be the fumigermin and fumihopaside BGCs. While the sporadic distribution patterns of these clusters could indicate their more accessory role within the ecology of *A. fumigatus*, further work is needed to verify that the observed pattern is not a consequence of the fumigermin or fumihopaside genes not always co-localizing within genomes—a pattern that would violate the assumptions made by antiSMASH and cBlaster in detecting the SM producing genes. Additionally, as the quality of the genomic sequences varies among the 264 published genomes, some false negatives in our dataset could be the result of highly fragmented assembled genomes and poor annotations.

Beyond the BGCs described in Lind et al. (2017) [117], we only found four BGCs that we confidently felt could be new clusters. Figure 6 visually depicts four GCFs considered novel in this research.

## 4. Discussion

Filamentous fungi, as diverse and adaptable organisms, are prolific producers of secondary metabolites, which contribute to their survival, reproduction, and interactions with other organisms in their environment [7]. Among the vast array of fungal species, *A. fumigatus* stands out as a significant pathogenic fungus with high mortality rates and increasing antifungal resistance, leading it to be placed on the WHO pathogen list of critical importance [119]. Several of its SMs have been shown to mediate various virulence traits of this pathogen [1,120]. Some—if not all—of these metabolites are also important in protection from various abiotic (e.g., UV, copper starvation) and biotic (e.g., microbial competitors) stressors [1,120]. The bioactivities of some *A. fumigatus* SMs have even led to practical applications, such as fumagillin treatment of the Nosema disease of honeybees [83]. Pyripyropene A and analogs, potent and selective sterol O-acyltransferase 2 (SOAT2)/acyl-coenzyme A:cholesterol aceyltransferase 2 (ACAT2) inhibitors, have promise for the treatment of atherosclerotic disease [74] and use as insecticides [121].

The linkage of 20 BGCs to specific products places *A. fumigatus*, along with *A. nidulans* [10], as one of the most thoroughly vetted species with regard to an understanding of its natural product potential. Not surprisingly, the first defined BGCs were those that correlated with commonly produced SMs. Several SMs were first chemically characterized, and the encoding BGC was identified by either gene similarity to known biosynthetic genes in other fungi (e.g., DHN melanin [33], gliotoxin [39], fumigaclavine [44,45], and fumitremorgin [49]), or by chemistry logic (e.g., if the SM was a peptide, then researchers looked for an NRPS), which led to BGC discovery for pseurotin A [55], helvolic acid [61], and fumiquinazoline [69].

At least five BGCs and their metabolites were characterized following a microarray study of the global SM regulator LaeA [99]. A transcriptional profile comparing WT, Δ*laeA*, and a complement control strain identified 13 BGCs that were downregulated in the deletion strain. Following this work, endocrocin [78], fumagillin [84] (and its entanglement with the pseurotin BGC) [59], hexadehydroastechrome [87], trypacidin [92,93], and fumisoquin BGCs [97,98] were linked to this product.

Advances in the characterization of other BGCs and their products took more effort. Two metabolites and their BGCs were identified via co-culture with *S. rapamycinicus*, fumicycline A (neosartoricin) [91] and fumigermin [105]. Xanthocillin synthesis is also induced in confrontations with a bacterium, *S. peucetius*, but its BGC was only identified via homology with known bacterial isocyanide synthases [100]. Several years later, two more isocyanides (fumivaline and fumicicolin), induced under copper starvation, were linked to the same BGC [106]. Finally, four metabolite/BGC pairings were only established using overexpression technologies, either in *A. fumigatus* itself or using the heterologous expression host *A. nidulans* [90]. These include neosartoricin (fumicycline A) [89,90], fumihopaside A [104], sphingofungin [107], and sartorypyrone [112].

Where does this leave the field of SM discovery in *A. fumigatus*? Most laboratory methods—use of mutants, different media, temperature shifts—to assess SM production in this fungus merely result in titer changes in the commonly expressed metabolites, as illustrated not only in this work (Figure 3 and Figure 4) but in many other studies [122,123,124,125].

Our analysis reveals a notably conserved secondary metabolome across *A. fumigatus* isolates. All but four of the BGCs that we identified either correlate with known chemical products (*n* = 22) or have been previously genetically characterized but remain unlinked to any natural product production (*n* = 18). Fully characterizing all BGCs with yet unidentified natural products will likely necessitate the use of overexpression technologies, whether endogenous or heterologous. However, the potential for discovering novel chemistries lies within the extensive array of putative BGCs that have yet to be chemically defined (Figure 5).

## Figures and Tables

**Figure 1 jof-10-00266-f001:**
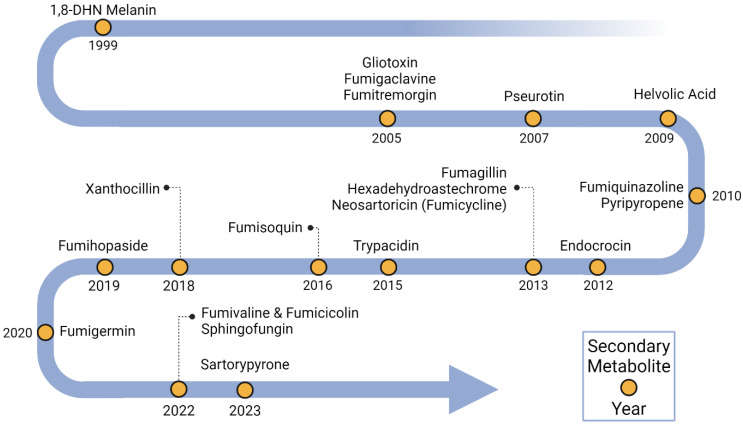
Timeline of BGC characterization in *Aspergillus fumigatus*. Yellow dots indicate secondary metabolites linked to BGCs and the years when their linkages were identified.

**Figure 2 jof-10-00266-f002:**
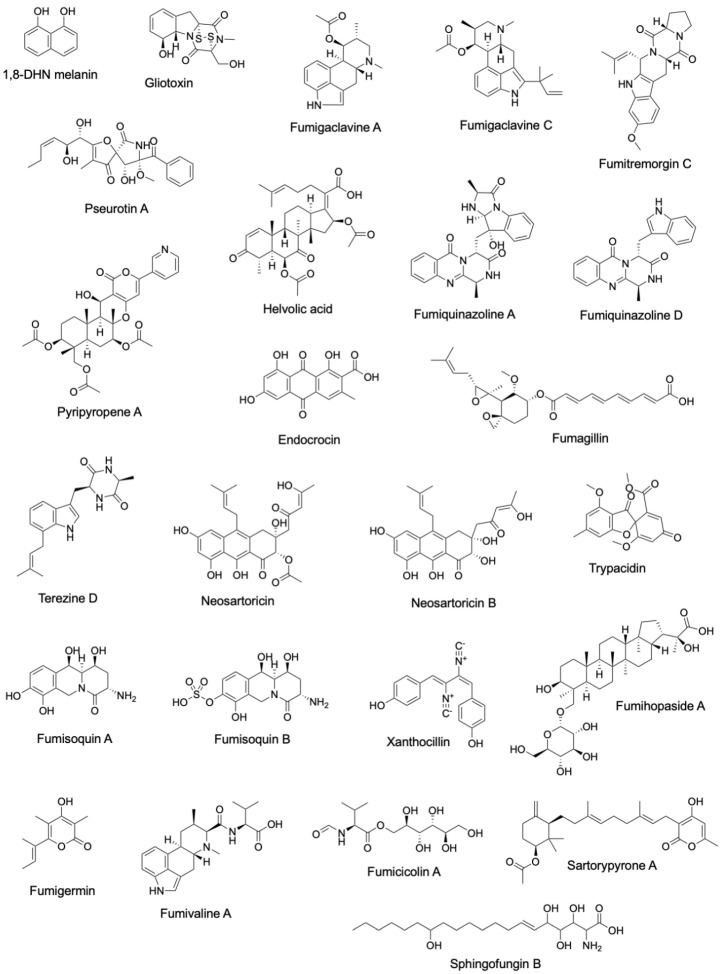
Structures of secondary metabolites linked to characterized *A. fumigatus* BGCs.

**Figure 3 jof-10-00266-f003:**
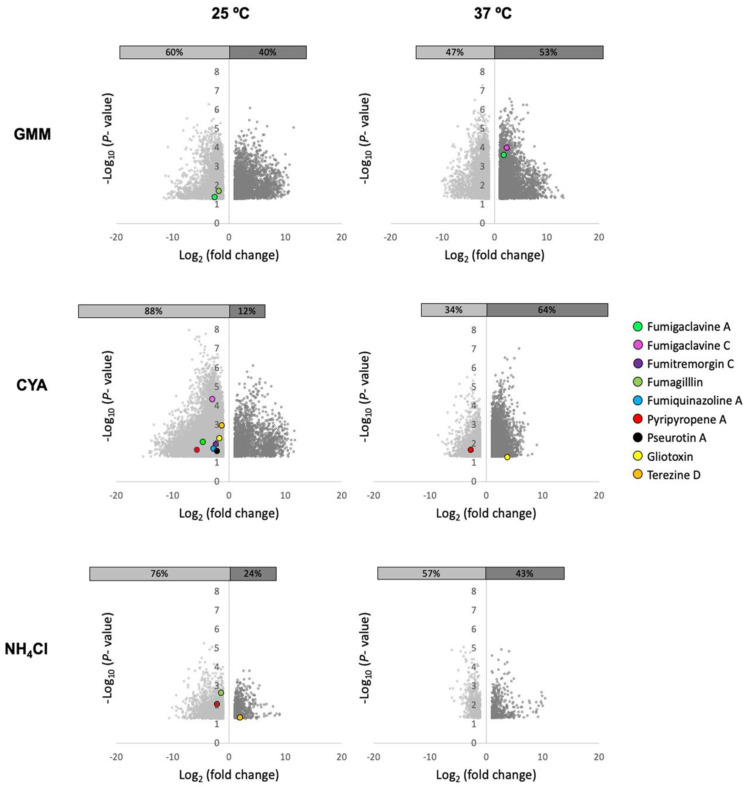
Volcano plots of significant features produced in *A. fumigatus* Δ*sirE* strain compared to WT when grown in diverse media and two temperatures. Co-regulation of numerous SMs in both strains varied based on the media compositions (GMM, CYA, NH_4_Cl) and temperatures (25 and 37 °C). Features with log_2_ fold change <1 and >1 and significant differences (*p* < 0.05) are shown.

**Figure 4 jof-10-00266-f004:**
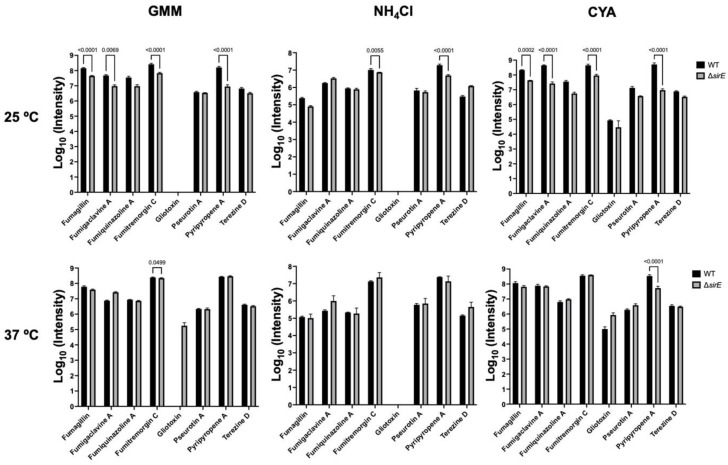
Intensities of *m*/*z* values of known SMs found in *A. flavus* WT and Δ*sirE* strains. Different metabolites were up- or downregulated in the Δ*sirE* strain compared to WT, depending on media and temperature.

**Figure 5 jof-10-00266-f005:**
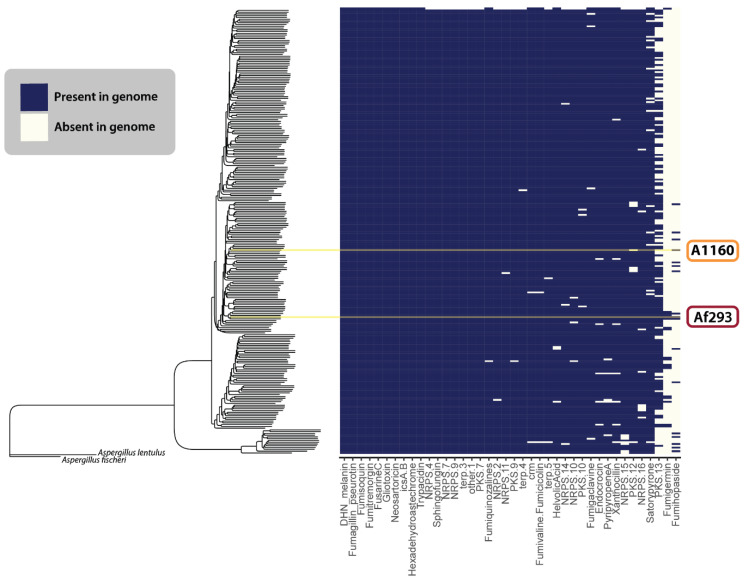
The BGCs of previously characterized secondary metabolism-producing genes in different isolates of *A. fumigatus*. The species tree estimation was generated using a coalescent model on a dataset of 700 single-copy ortholog gene trees. The heatmap shows the presence (blue) or absence (cream) of various secondary metabolite GCFs across 264 *A. fumigatus* isolates. The yellow lines indicate Af293 and A1160.

**Figure 6 jof-10-00266-f006:**
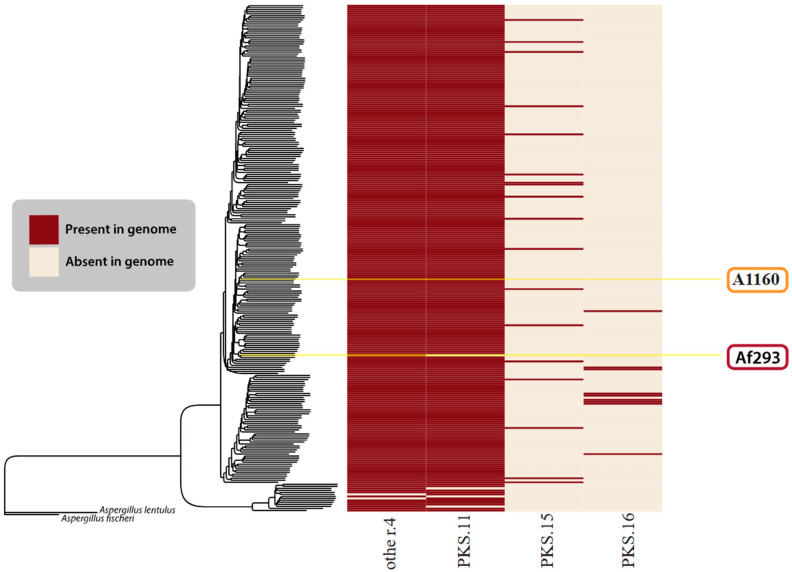
Novel BGCs identified in this study. The species tree is identical to that in Figure 5. The heatmap shows the presence (red) or absence (cream) of all novel/unknown GCFs found in the *A. fumigatus* isolates.

## Data Availability

Data are contained within the article and Appendix A.

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
