# Peer review of "A Timeline of Biosynthetic Gene Cluster Discovery in Aspergillus fumigatus: From Characterization to Future Perspectives"

_jof, 2024, doi:10.3390/jof10040266_

Round 1

Reviewer 1 Report

Professor Keller et al. present a very nice and comprehensive overview of the historical journey and current state of research on NPs derived from A. fumigatus. It is somewhat a non-traditional article, as it mostly presents the historical review along with bioinformatics analysis of Biosynthetic Gene Clusters (BGCs) from A. fumigatus. Overall, the paper is very nicely written and could be published with very minor revisions.

1. Line 78 -20 0C, should be a space in between 

2. Line 372 A fumigatus italic 

3. I would suggest the authors include chemical structures for the NPs from A. fumigatus.

4. In terms of the absence of some BGCs in some strains of A. fumigatus, would it be possible that this is because of gaps between contigs, as most of the sequencing was done by second-generation sequencing?

Author Response

Major comments

Professor Keller et al. present a very nice and comprehensive overview of the historical journey and current state of research on NPs derived from A. fumigatus. It is somewhat a non-traditional article, as it mostly presents the historical review along with bioinformatics analysis of Biosynthetic Gene Clusters (BGCs) from A. fumigatus. Overall, the paper is very nicely written and could be published with very minor revisions.

  1. We appreciate your feedback. We are glad to know that you interpreted the manuscript as we intended.

Detail comments

  1. Line 78 -20 0C, should be a space in between

A: Thank you for pointing out this typo in the manuscript. The modified part is marked in red in the manuscript.

  1. Line 372 A fumigatus italic 

A: We appreciate your help in identifying the typo within the manuscript. The modified part is marked in red in the manuscript.

  1. I would suggest the authors include chemical structures for the NPs from A. fumigatus.
  2. Thank you for your feedback. We totally agree with your suggestion and have incorporated it into the revised manuscript by introducing a new Figure 2.
  3. In terms of the absence of some BGCs in some strains of A. fumigatus, would it be possible that this is because of gaps between contigs, as most of the sequencing was done by second-generation sequencing?
  4. We agree with the reviewer that on an individual genome level, the absence of a BGC could be attributed to gaps between contigs, sequencing errors, annotation oversights, or the physical separation of contigs containing BGCs. These factors pose a significant challenge when dealing with uncharacterized BGCs. To mitigate such impacts, our methodology for the known clusters allowed for the possibility of BGCs being fragmented across different contigs. Nonetheless, we recognize that incomplete sequencing or improper assembly of regions harboring BGCs can lead to false negatives. We agree that this consideration is crucial and have thus incorporated an additional sentence in our review to address this concern.

Reviewer 2 Report

This is a very interesting and readable chronological account of the isolation and characterisation of biosynthetic gene clusters from a single organism. It will be of general interest to a wide variety of natural product researchers and I can happily recommend publication. Just a few comments.

1. As a natural product chemist I found the lack of actual  chemical structures both frustrating and annoying despite being familiar with many of them. They vary enormously in structure and biogenesis and while structure is alluded to in the discussion, why not just show them!

2. Minor - line 38 should be "reknowned" and l41 "any one".

3. Line 233 - needs  rewriting. Trypacidin is NOT an anthraquinone, it is a grisan but is biosynthesised via an AQ intermediate.

4. line 245. Wrong structure for isocyanide - should be -N+triple bond C-

1. As a natural product chemist I found the lack of actual  chemical structures both frustrating and annoying despite being familiar with many of them. They vary enormously in structure and biogenesis and while structure is alluded to in the discussion, why not just show them!

2. Minor - line 38 should be "reknowned" and l41 "any one".

3. Line 233 - needs  rewriting. Trypacidin is NOT an anthraquinone, it is a grisan but is biosynthesised via an AQ intermediate.

4. line 245. Wrong structure for isocyanide - should be -N+triple bond C-

Author Response

Major comments

This is a very interesting and readable chronological account of the isolation and characterisation of biosynthetic gene clusters from a single organism. It will be of general interest to a wide variety of natural product researchers and I can happily recommend publication. Just a few comments.

  1. We appreciate your feedback. We're pleased to hear that you understood the manuscript as we intended.

Detail comments

  1. As a natural product chemist I found the lack of actual chemical structures both frustrating and annoying despite being familiar with many of them. They vary enormously in structure and biogenesis and while structure is alluded to in the discussion, why not just show them!
  2. Thank you for your feedback. We totally agree with your opinion and have incorporated structures into the revised manuscript by introducing a new Figure 2.
  3. Minor - line 38 should be "reknowned" and l41 "any one".
  4. Thank you for pointing out these typos in the manuscript. We modified the terms "renowned" and "the" to convey the intended meaning accurately. The modified part is marked in red in the manuscript.
  5. Line 233 - needs rewriting. Trypacidin is NOT an anthraquinone, it is a grisan but is biosynthesised via an AQ intermediate.
  6. We appreciate your identifying this mistake. We agree with your opinion that trypacidin is grisan not anthraquinone. We’ve searched for the paper to corroborate this information. and we found a paper (Turner, W. B. (1965). 1232. The production of trypacidin and monomethylsulochrin by Aspergillus fumigatus. Journal of the Chemical Society (Resumed), 6658-6659.) proving that trypacidin is a spirocoumaranone. As grisan belongs to the spirocoumaranone class, we changed the part you point out by referencing spirocoumaranone. The modified part is marked in red in the manuscript.
  7. line 245. Wrong structure for isocyanide - should be -N+triple bond C-
  8. We appreciate your help in identifying the typo within the manuscript. The modified part is marked in red in the manuscript.